# Two-dimensional tessellation by molecular tiles constructed from halogen–halogen and halogen–metal networks

Fang Cheng[1], Xue-Jun Wu[2], Zhixin Hu [3], Xuefeng Lu[1], Zijing Ding[4], Yan Shao[5], Hai Xu[1,4], Wei Ji [6], Jishan Wu [1] & Kian Ping Loh [1,4]

Molecular tessellations are often discovered serendipitously, and the mechanisms by which specific molecules can be tiled seamlessly to form periodic tessellation remain unclear. Fabrication of molecular tessellation with higher symmetry compared with traditional Bravais lattices promises potential applications as photonic crystals. Here, we demonstrate that highly complex tessellation can be constructed on Au(111) from a single molecular building block, hexakis(4-iodophenyl)benzene (HPBI). HPBI gives rise to two self-assembly phases on Au(111) that possess the same geometric symmetry but different packing densities, on account of the presence of halogen-bonded and halogen–metal coordinated networks. Sub-domains of these phases with self-similarity serve as tiles in the periodic tessellations to express polygons consisting of parallelograms and two types of triangles. Our work highlights the important principle of constructing multiple phases with self-similarity from a single building block, which may constitute a new route to construct complex tessellations.

[1] Department of Chemistry, National University of Singapore, Singapore 117543, Singapore. [2] State Key Laboratory of Coordination Chemistry, School of Chemistry and Chemical Engineering, Nanjing University, 210023 Nanjing, China. [3] Center for Joint Quantum Studies and Department of Physics, Institute of Science, Tianjin University, 300350 Tianjin, China. [4] Centre for Advanced 2D Materials and Graphene Research Centre, National University of Singapore, Singapore 117546, Singapore. [5] Institute of Physics & University of Chinese Academy of Sciences, Chinese Academy of Sciences, 100190 Beijing, China. [6] Department of Physics and Beijing Key Laboratory of Optoelectronic Functional Materials & Micro-Nano Devices, Renmin University of China, 100872 Beijing, China. These authors contributed equally: Fang Cheng, Xue-Jun Wu, Zhixin Hu. Correspondence and requests for materials should be addressed to K.P.L. (email: chmlohkp@nus.edu.sg)

Two-dimensional (2D) tessellation involves the tiling of a plane using one or more closed shapes without the formation of overlaps or gaps. The idea of tessellation, which is of great importance in aesthetics[1], mathematics[2,3], chemistry[4], and molecular science[5,6], can be traced back to building decorations used by the Sumerians in ancient times[7]. Archimedean tiling (AT), an archetypical tessellate form, is based on the tessellation of regular polygons and can be classified into three regular tiling types and eight semi-regular tiling types[8]; the former relies on basic tiling of one specific polygon (squares, triangles, or hexagons), while the latter consists of tessellations of more than one type of regular polygons. Compared with regular ATs, semi-regular ATs possess intriguing photonic[9–13] and diffusion[14] properties. Owing to their higher rotational symmetry than the traditional Bravais lattices, semi-regular ATs have been reported to exhibit isotropic photonic bandgaps, and the isotropy is enhanced with the increasing complexity of the structures[9–11]. Although it is trivial to demonstrate regular AT at the atomic level (e.g. the square lattice of Cu(100) and the honeycomb structure of graphene), it remains challenging to construct semi-regular ATs at the supramolecular level.

Supramolecular chemistry[15], which relies on spontaneous and reversible non-covalent interactions[16–22], is attractive as it enables highly versatile fabrication of molecular architectures on surfaces. In the past decade, great effort has been devoted to the development of semi-regular AT[23–27] and complex tilings[28–34] on surfaces using self-assembly approaches. For instance, semi-regular trihexagonal tiling, also known as the Kagomé lattice[35], is frequently realized in two-dimensional (2D) self-assembly patterns[23–26]. To explore much more complex tessellations, Barth and co-workers reported rare earth metal-directed self-assembly structures, which were stabilized through metal–organic coordination interaction[36,37]. Taking advantage of the versatility of lanthanide coordination chemistry, a variety of coordination molecular motifs were constructed by tuning the metal-to-organic molecule ratio. Random tiling of the various molecular

motifs generates quasicrystalline structures[36]. Non-planar rubrene molecule tessellation has also been reported; in addition to two ordered phases, a non-periodic phase was constructed via the random mixing of pentagonal, hexagonal, and heptagonal units[38]. The ability to form tessellated structures implies certain self-similarity and hierarchical order in the building blocks, but this organization can be expressed in different ways in multi-nuclear nodes, which are moderated by steric constraint and molecule–substrate interaction. Compared to the non-periodic tessellation, quasicrystal and periodic tessellations with high degree of rotational symmetry are promising candidates for 2D photonic crystals[10]. However, the construction of periodic higher-order tessellation from two-ordered phases has been reported only rarely[39].

Herein hexakis(4-iodophenyl)benzene (HPBI), a $D_{6h}$ symmetric molecule, is used as a building block to fabricate two periodic supramolecular phase types on Au(111). These self-assembly patterns have the same lattice symmetry but different packing densities; the variation in packing density is attributed to networks formed by halogen bonds[40,41] and halogen–Au coordination, as verified via scanning tunnelling microscopy (STM) and density function theory (DFT) calculations. Notably, the self-similarity of these two phases favours interweaving between the phases to form higher-order supramolecular phases; this mechanism constitutes a new method for constructing complex 2D tessellations through the interplay between intermolecular and molecule–substrate interactions[42,43].

## Results

**Formation of two supramolecular phases.** The mesogenic molecular building block, namely HPBI[44,45] (synthesis details can be found in Supplementary Method), is a nonplanar molecule composed of a central benzene ring flanked by six out-of-plane *p*-iodophenyl groups with a lateral width of about 1.5 nm (Fig. 1a). Deposition of a submonolayer of HPBI on Au(111), followed by

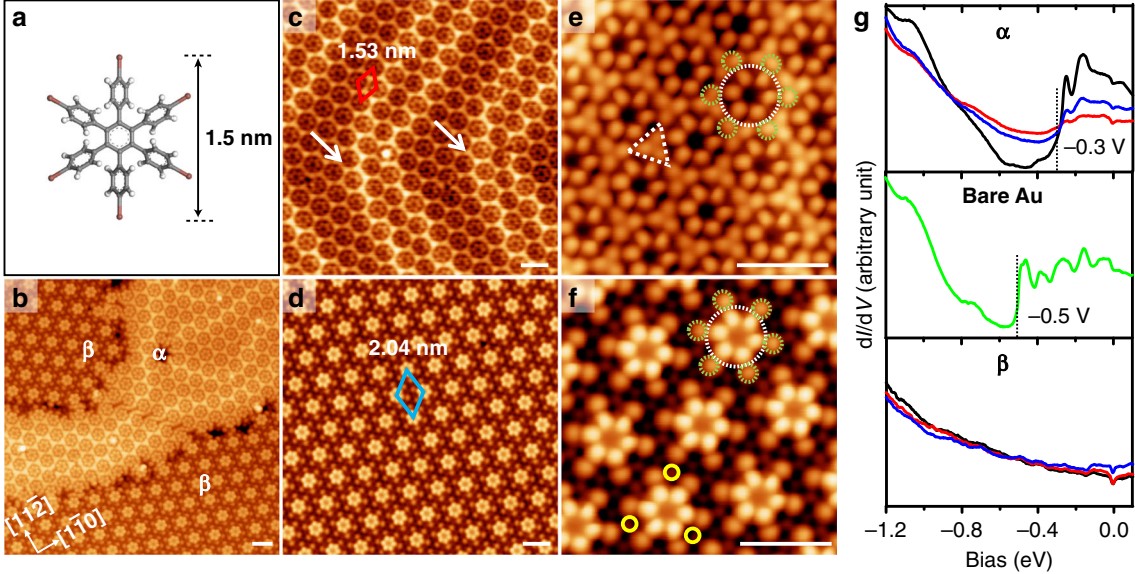

**Fig. 1** The two-ordered self-assembly phases of HPBI on Au(111). **a** The chemical structure of HPBI. Carbon, grey; iodine, red; hydrogen, white. **b** Large-scale STM image of the coexisting α and β phases. The crystalline axes of the Au substrate are labelled. **c**, **d** STM images of the pure α (**c**) and β (**d**) phase domains. The white arrows in **c** designate the herringbone structure in the α phase; the α and β phase unit cells are represented by red and blue rhombi, respectively. **e**, **f** Magnified STM images of the α (**e**) and β (**f**) phases. HPBI molecules are labelled by the dashed white and green circles. The dashed white triangle in **e** highlights the I–I trimer. The solid yellow circles in **f** highlight the adatoms. **g** STS spectra acquired for the α (upper panel) and β (lower panel) phases and bare Au surface (middle panel). The black, blue, and red curves represent the spectra taken at the I atom, peripheral phenyl, and centre phenyl, respectively. Scale bars in all STM images: 2 nm

subsequent cooling to 0.4 K, generates two self-assembly phases with the same symmetry but different packing densities (Fig. 1b). The representative STM images of both phases (Fig. 1c, d) reveal lattice constants (red and blue unit cells) of 1.53 and 2.04 nm, giving rise to α and β phases with high (0.49 molecule nm$^{-2}$) and low (0.28 molecule nm$^{-2}$) packing densities, respectively. A herringbone-like reconstruction is clearly visible in the α phase (white arrows in Fig. 1c) but is not detected in β phase, which indicates different molecule–substrate interactions in the two phases.

These two phases are detailed in Fig. 1e, f. A single HPBI molecule consists of a flower-shaped feature with six-fold symmetry, which is consistent with the $C_6$ symmetry of the molecule. The central lobes (dashed white circle) are assigned to phenyl groups, where the contrast between the peripheral and central phenyl groups results from the nonplanar structure of HPBI; the peripheral protrusions (dashed green circles) are I atoms connected to the phenyl groups through C–I bonds. In the α phase (Fig. 1e), three I atoms from the adjacent molecules form a triangular arrangement called a halogen-3 synthon (i.e. the dashed white triangle), which is highly polarizable; owing to this polarizability, I atoms can develop σ-holes that enable them to form halogen bonds with nucleophilic I neighbours; similar halogen-bonded triangles have been reported previously[46–48]. The β phase can be considered as a derivative of the α phase, albeit with an enlarged intermolecular distance. Figure 1f reveals additional faint features adjacent to the I atoms (highlighted by solid yellow circles). These faint features should not be dissociated I atoms, as no dissociated aryl groups were found on the surface, which implies that HPBI was deposited intact herein. Previous studies have reported that the removal of the herringbone reconstruction is accompanied by the appearance of Au adatoms, and the formation of Au adatoms-mediated self-assembly[49–51]. Considering that the herringbone reconstruction is lifted in the β phase, it is reasonable to propose that the molecular network is stabilized by coordination between I atoms and Au adatoms.

The herringbone reconstruction on Au(111) is a manifestation of excess surface stress; the lifting of the reconstruction is commonly attributed to stress relief mechanism induced by molecular adsorbates[52–54]. Our molecular coverage experiments involving HPBI show that α phase first emerges at low molecular coverage (Fig. 2a), while the β phase appears as coverage increases and coexists with α phase (Fig. 2b). We speculate that, when the coverage of HPBI molecules reaches a critical threshold on the surface to affect the strain or charge density in one localized domain, the herringbone reconstruction is lifted and Au adatoms

are ejected, leading to a phase transition to the β phase where Au adatoms participate in Au–I coordination network. The phase evolution of our supramolecular network is convoluted with the coverage-dependent evolution of surface stress on the Au(111) substrate.

Scanning tunnelling spectroscopy (STS) was used at various molecular locations to investigate the interactions between the molecules and the Au substrate. As shown in Fig. 1g, d$I$/d$V$ spectra of α phase HPBI display a step-like feature at –0.3 eV, while the surface state of the pristine Au(111) (middle panel) shows a characteristic step at –0.5 eV. The 0.2 eV upward shift is attributed to adsorption-induced modification of the surface state[55–57]. However, STS spectra of the β phase molecules show no evidence of the Au(111) surface state, this may be due to structural change or charge rearrangement[58,59] at the Au–HPBI interface in the β phase, which is consistent with the appearance of Au adatoms on the substrate.

DFT calculations were performed to provide insight into the bonding interactions in both phases. The α phase model (Fig. 3a) shows that I atoms, which are adsorbed close to the Au top sites, form triple I–I halogen bonds (dashed black lines). The simulated STM image (Fig. 3c) shows excellent agreement with our experimental results (Fig. 1e). Figure 3b shows the fully relaxed β phase model, in which one Au adatom appears for every three neighbouring I atoms, consistent with the STM results herein. In addition, the calculated unit cell lattice (2.07 nm, blue rhombus in Fig. 3d) agrees well with the experimental lattice constant (2.04 nm) determined for the β phase.

Frontier molecular orbitals were calculated to analyse intermolecular interactions. For the α phase, the highest occupied molecular orbitals (HOMO) and lowest unoccupied molecular orbitals (LUMO) have their projections on the I atoms at –1.1 eV and 2.3 eV, respectively (upper panel of Supplementary Fig. 1). Figure 3e shows that the wave functions of these states overlap within the I trimer region, which indicates I–I bonding. On the other hand, the distances between adjacent I atoms are much larger in the β phase due to coordination with Au adatoms to form I–Au–I interactions, as indicated by the plotted wavefunctions of HOMO (–1.5 V) and LUMO (2.9 V) shown in Fig. 3f. The three-centre orbital hybridization has been also reported in N–Au–N and biphenyl trimer systems previously[60,61]. Based on the structural models in Fig. 3a, b, the binding energies of the α and β phases are calculated to be –4.17 and –4.08 eV, respectively. The close similarity in binding energy between the two phases explains their coexistence in our experiments.

Although HPBI is an achiral molecule in the gas phase, two chiral isomers can be generated upon adsorption on surface as the tilted peripheral phenyls adopt clockwise or counter-clockwise rotations with respect to the axis perpendicular to the centre phenyl (Supplementary Fig. 2a). The chiral isomers can be resolved in high-resolution STM images, which show that the pure enantiomer usually appears in the α phase, while the racemic mixture is commonly observed in the β phase (Supplementary Fig. 2b and c). Beyond single molecules, the molecular arrangements in the α and β phases also exhibit chirality (Supplementary Fig. 3), which can be identified by the relative rotation between the molecule orientations and supramolecular organizations.

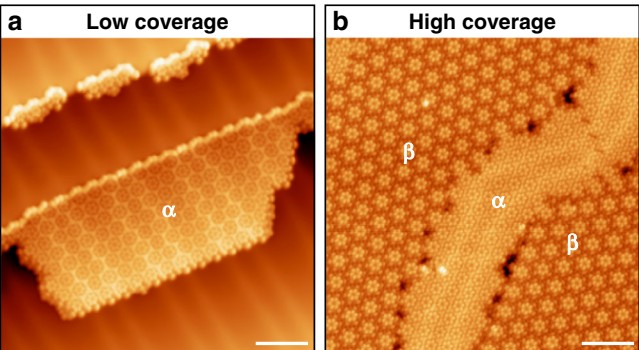

**Fig. 2** Coverage-dependent evolution of α and β phases. **a** STM image taken after the initial deposition of HPBI at low surface coverage. Molecules assemble to form close-packed α phase. **b** STM image taken at higher surface coverage of HPBI. Less compact β phase starts to appear on surface coexisting with α phase. Scale bar in all STM images: 5 nm

**Higher-order interwoven tessellation.** The α and β phase hexagonal arrangements can be viewed as geometric expressions of regular AT (3$^6$), where molecule centres are placed at the vertices of the polygons. Figure 4a shows a schematic illustration in which the distinct intermolecular distances connecting the adjacent vertices in the α and β phases are labelled by red and blue lines,

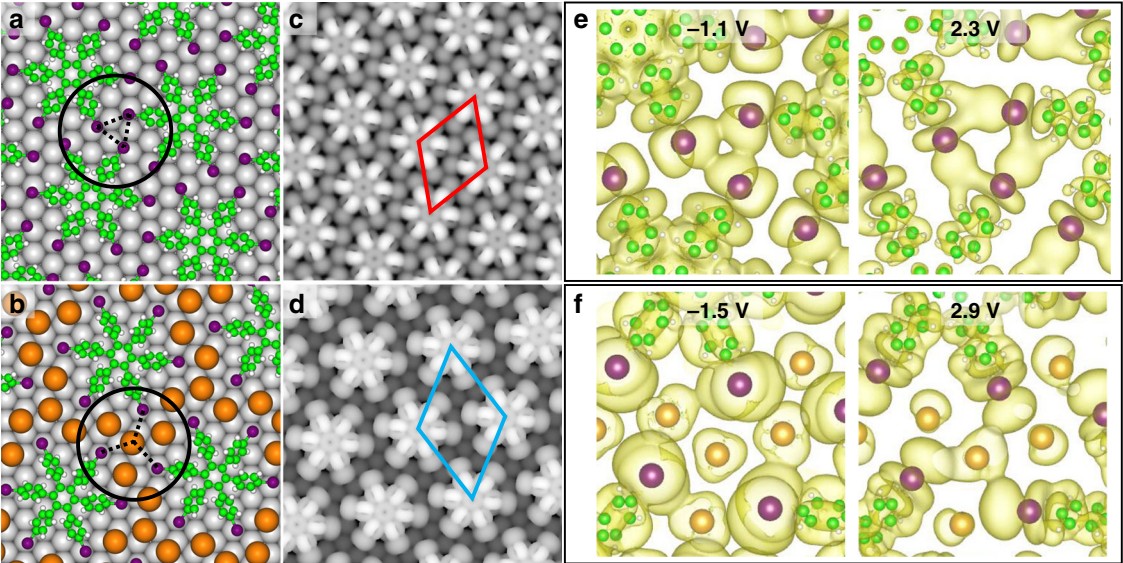

**Fig. 3** DFT calculations of the self-assembled α and β phases. **a, b** Calculated structural models for the α (**a**) and β (**b**) phases. **c, d** STM simulations of the α (**c**) and β (**d**) phases based on the structural models in **a**, **b**. **e, f** Real-space visualized wavefunctions of HOMO and LUMO in the α (**e**) and β (**f**) phases. Colour definitions: carbon, green; iodine, purple; Au adatom, orange

respectively. One might ask whether these two phases can interweave to form a more complex architecture. Previous reports typically show formation of non-periodic structures when two or more ordered phases co-exist on the substrate[38]. In this work, in addition to the large-scale, phase-separated α and β phases, inhomogeneously mixed phases (Supplementary Figs. 4 and 5) are indeed observed in selected local areas as the HPBI coverage was increased above 0.55 ML (1 ML is defined as one monolayer of HPBI in α phase). Most intriguingly, three types of ordered mixed-phase structures (named A, B, and C) are discovered under high coverage regime; STM images and corresponding fast Fourier transform (FFT) patterns of these images show periodic arrangements of close-packed clusters with six-fold symmetry, as shown in Fig. 4b–d and insets therein. Three molecular cluster types can be resolved in these ordered structures, including monomers, triangular-shaped trimers, and hexagonal-shaped heptamers; the inter-molecular distances within the trimer and heptamer are equal to those in the α phase. Structure A is based on a monomer and a trimer, while structure B is based on a heptamer; structure C is based on a trimer and a heptamer. Interestingly, the distances between adjacent molecules located at the separated clusters (blue lines in Fig. 4b–d) is about 2.04 nm, equal to those in the β phase. Therefore, these complex structures can be interpreted as interwoven tessellations of the α and β phases. For a clearer view, the packing topologies were sketched and overlain on the STM images (Fig. 4b–d); the red and blue lines label intermolecular distances similar to those in the α and β phases, respectively. Careful comparison reveals that structures A and B can be interconverted through a simple exchange of red and blue lines. Structure C features a more complicated architecture containing units, which appear in structures A and B.

Enlarged views of STM images (Fig. 4e–g) reveal the detailed structures around the tessellation vertices. In structure A, the polygons that cross at the interwoven vertex (black circle in Fig. 4e) consist of one red triangle, two parallelograms, and two blue triangles. The red triangle is perfectly aligned with one of the blue triangles (highlighted by dashed white lines), leading to 60° and 120° interior angles in the parallelogram. The angular sum of the polygons involved is therefore 360°, resulting in perfect tessellation. The tile representation in Fig. 4h highlights the interwoven vertex, which is consistent with the observations in

Fig. 4e. According to the nomenclature defined by Grünbaum and Shephard[2], the lattice can be named using a set of integers in the form $(n_1^{\alpha 1}.n_2^{\alpha 2}.\cdots)$ to represent a vertex surrounded by polygons in a clockwise sequence; $n_i$ represents the number of sides on the polygon, and $\alpha i$ denotes the number of adjacent polygons of the same type[14]. Since we have two triangle types with different side lengths, we differentiate them as $3_\alpha$ and $3_\beta$. Therefore, structure A can be described as tiling $(3_\alpha.4.3_\beta^2.4)$. Apparently, this packing geometry has a chiral structure. The STM images of two enantiomorphous domains are shown in Supplementary Fig. 6. Because structure B can be interconverted to structure A by exchanging red and blue lines, it can be described as tiling $(3_\alpha^2.4.3_\beta.4)$ (see tile representation in Fig. 4i). Structure C contains two types of interwoven vertex configurations (Fig. 4j). As shown in Fig. 4g, the vertex marked by the yellow circle has an arrangement identical to that in structure B (Fig. 4f), while the vertex marked by the black circle is the chiral isomer of that shown in structure A (Fig. 4e). This complex configuration can be expressed as tiling $(3_\alpha.4.3_\beta^2.4)$; $(3_\alpha^2.4.3_\beta.4)$.

These complex tessellations can be regarded as two different phases interweaving in multiple ways. Although 2D molecular tessellation involving multiple polygon types has been demonstrated previously[27,36–38], most studies reported construction from one specific assembly phase. In our case, owing to the co-existence of two distinct lattice constants originating from two phases, irregular polygons (parallelograms) are incorporated in the tessellation when two sub-domains with different lattices cross at a common vertex. Despite this complexity, the three higher-order structures maintain the six-fold symmetry of the α and β phases. Previous theoretical investigations have demonstrated that balanced entropy and enthalpy plays an important role in the formation of quasicrystalline and related complex structures[62,63]. From an entropic view point, random mixing of α and β sub-domains should occur; however, the highly oriented packing of α and β sub-domains with the gold substrate indicates considerable molecule–substrate interaction, which provide the enthalpy necessary for the generation of the periodic interweaving architectures. The formation of the manifold complex tessellations is most likely due to the competition between halogen–halogen, halogen–metal, and molecule–substrate interactions.

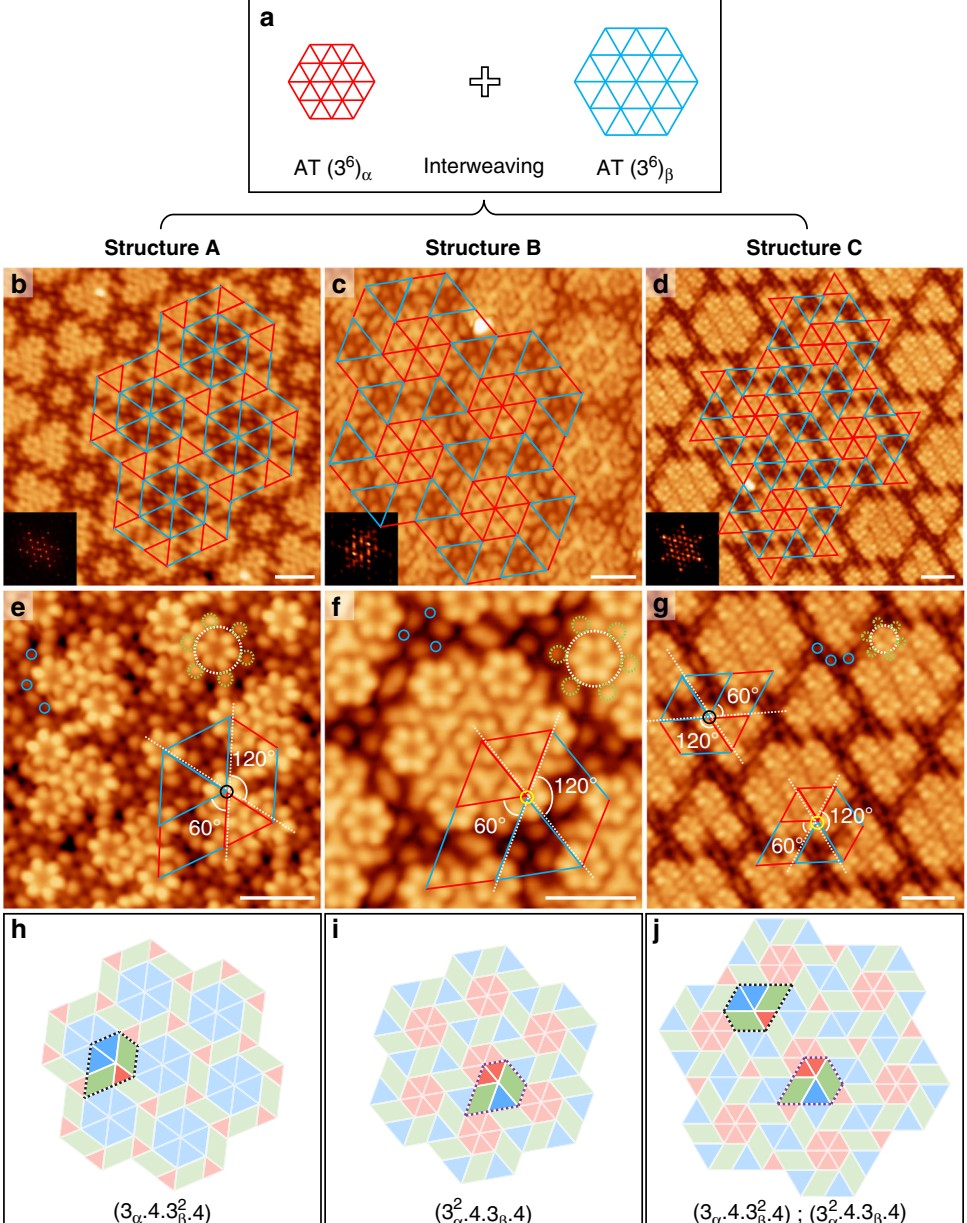

**Fig. 4** Periodic higher-order interwoven tessellations. **a** Schematic illustration of the AT in the α and β phases. **b–d** STM overviews of the three types of ordered interwoven structures with superimposed packing topologies. Insets: corresponding FFT patterns. **e–g** High-resolution STM images of structures A (**e**), B (**f**), and C (**g**). HPBI molecules are labelled by the dashed white and green circles, and the Au adatoms are labelled by blue circles. The interwoven vertices are marked by black and yellow circles. **h–j** Schematic tiling representations of structures A (**h**), B (**i**), and C (**j**) with highlighted interwoven vertex configurations. Triangles are red and blue, and parallelograms are green. In all images, intermolecular distances similar to those in the α and β phases are denoted by red and blue lines, respectively. Scale bars in all STM images: 2 nm

In order to determine whether the self-assembled network facilitates direct C–C coupling reactions, Ullmann coupling is initiated by annealing the sample containing various self-assembled phases to 200–400 °C, followed by cooling down and measuring at 0.4 K. Periodic complex tessellations are not detected after annealing, instead, triangle-shaped oligomers (Fig. 5a) are formed, which inherit the parental topology of the self-assembled triangular-shaped α domains. Enhanced local density of state observed at the junctions between molecules (Fig. 5a) indicates chemical bond formation (schematic model, Fig. 5d). Owing to molecular rotation and diffusion at elevated temperature, oligomers with different structures are also detected in our experiments. For example, dimer and trimer chains (STM

images in Fig. 5b, c, schematic models in Fig. 5e, f) are observed. The distances between adjacent dimers or trimers (marked by blue lines in Fig. 5b, c) are the same as the intermolecular distance observed in β phase, which indicates that the surface reaction is guided by the pre-assembled structure. This reveals that on-surface reactions can be controlled by the pre-packed molecular assembly, which could constitute a promising strategy for the synthesis of C–C-coupled molecular clusters.

## Discussion
We have demonstrated that ordered supramolecular architecture can be constructed from the tessellation of high- and low-density

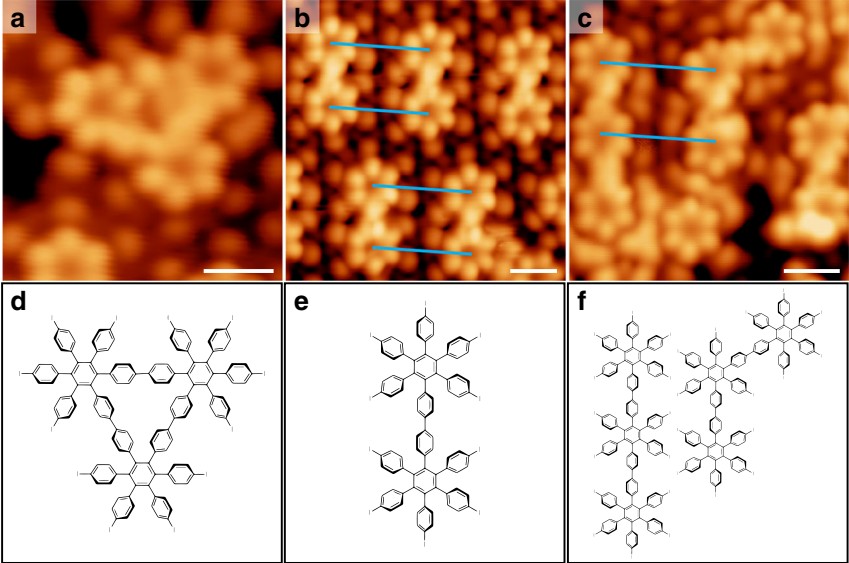

**Fig. 5** Ullmann reaction of HPBI on Au(111). **a** STM image of a triangular oligomer formed from three HPBI molecules. **b** STM image of linear oligomers formed from two HPBI molecules. **c** STM image of linear and angled oligomers formed from three HPBI molecules. The distances between the adjacent oligomer chains are the same as the intermolecular distances in the β phase, as indicated by the blue lines. **d–f** The corresponding structure models of STM images in **a–c**. Scale bar in all STM images: 1 nm

molecular phases. The two molecular phases arise from distinct intermolecular and molecule–substrate interactions; the high-density phase is constructed from halogen bonds, while the low-density phase is formed via a coordinated halogen–Au network. The evolution of these molecular phases are convoluted with strain-relieving mechanism on the reconstructed domains of Au (111). The self-similarity of the two molecular phases allows clusters from each to tessellate and form high-order supramolecular networks. This approach opens up new route to construct complex surface tessellation by considering the symmetry of building block and substrate, as well as by introducing multimode interactions. In addition, the complex tessellations in our work may provide new insights for understanding self-organized systems in biology and nanotechnology[64].

## Methods

**Sample preparations**. All experiments were carried out in an ultrahigh vacuum system (base pressure <$3.0 \times 10^{-10}$ mbar) equipped with a Unisoku low-temperature STM. A single Au(111) crystal was cleaned by repeated cycles of Ar$^+$ sputtering (1.0 keV) followed by annealing at 600 °C, until no contamination could be detected by STM. Before use, the HPBI powder was outgassed at 300 °C for several hours. HPBI molecules were then deposited from a Knudsen cell at 420 °C; the substrate was kept at room temperature during molecule deposition.

**STM/STS experiments**. After the deposition of HPBI on the Au(111) substrate, the sample was transferred in situ to the analysis chamber equipped with a low-temperature STM. All STM and STS measurements were conducted at 0.4 K with electro-chemically etched tungsten tips. The STM images were collected under constant-current mode. The piezo was calibrated using an HOPG sample. The STS point spectra (d$I$/d$V$) were measured using standard lock-in techniques with a voltage modulation of 10 mV and frequency of 987 Hz. Before real-sample STS measurements, the tip was calibrated on a clean Au(111) sample or the bare Au area on a real sample. STM/STS data were analysed and rendered using the WSxM software.

**Theoretical calculations**. Calculations were performed using the Perdew–Burke–Ernzerhof function (for the exchange-correlation function)[65], the projector augmented method[66,67], and a plane-wave basis set, as implemented in the Vienna ab initio simulation package[68]; a semi-empirical dispersion correction (DFT-D3) proposed by Grimme et al. was used in the simulation[69]. The energy cutoff for the plane-wave basis was set to 400 eV. The optimized lattice parameter for the bulk Au was $a = 4.097$ Å. Slab models were used to simulate the Au(111)

surface. The slab was constructed with 5 layers of Au atoms and a vacuum space of 20 Å in the $z$ direction. Only gamma points were used to sample the first Brillouin zone of the supercell. All atoms except for the bottom layer were fully relaxed until the residual force for each atom was <0.02 eV Å$^{-1}$. STM simulation was performed using the Tersoff–Hamann approximation[70]. The charge densities on the surface within a specific energy range, which are selected to match with the experimental bias, are calculated pixel by pixel. The constant charge density contour is generated as the simulated constant-current image, where the intensity of a specific pixel is determined by its vertical height. The interaction between tip and surface is ignored in the calculations.

## Data availability
All data are available from the authors upon reasonable request.

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

## Acknowledgements

K.P.L. acknowledges NRF-CRP grant "Two Dimensional Covalent Organic Framework: Synthesis and Applications". Grant number NRF-CRP16-2015-02, funded by National Research Foundation, Prime Minister's Office, Singapore. The authors gratefully acknowledge funding support from MOE Tier 3 programme (MOE2014-T3-1-004), the

National Natural Science Foundation of China (Grant Nos. 91433103, 11622437, 11804247), the Fundamental Research Funds for the Central Universities of China (Grant Nos. 16XNLQ01 (RUC), 020514380162), and Strategic Priority Research Program of Chinese Academy of Sciences (Grant No. XDB30000000). Calculations were performed at the Physics Lab of High-Performance Computing of Renmin University of China and the Shanghai Supercomputer Center.

## Author contributions

F.C., X.-J.W., and K.P.L. conceived the idea and designed the experiments. F.C., Y.S., and H.X. performed the STM and STS measurements. Z.H. and W.J. carried out the theoretical calculations. X.L. synthesized the molecule precursor under the supervision of J.W. F.C., X.-J.W., Z.H., Z.D., W.J., and K.P.L. analysed the data. F.C., X.-J.W., and K.P.L. wrote the paper with input from all the authors.

## Additional information

**Competing interests:** The authors declare no competing interests.

