## [Peer Review File · Nature Communications]

Reviewers' comments:

Reviewer #1 (Remarks to the Author):

This is a very nice piece of work with some beautiful structures which are described in a clear manner. Such complex structures showing periodic tessellations of different tiles are rare and of significant interest. As the authors note the controlled assembly of such structures should have interest for photonic materials. Of course the applications are beyond the scope of this paper but it is important to develop the fundamental underpinning science. The authors use a combination of high quality scanning probe studies supported by calculations to provide an evaluation of the observed structures.

Overall I recommend that the results are of significant interest but that a number of points need to be considered and answered.

The authors comment that "The phase evolution of our supramolecular network is convoluted with the coverage-dependent evolution of surface stress on the Au (111) substrate. A detailed investigation is beyond the scope of the current study,"

Although I agree that there is a large amount of work to be able to determine a full phase diagram for the system and that is indeed beyond the scope of this study. However it would be good if the authors could clarify some points. The molecules are deposited onto the substrate at room temperature, presumably 298K. What is the structure of the assembly at that temperature, prior to cooling to 0.4 K? This should be an easy experiment to do, if the authors haven't already done it, and this would give an indication of the structure at elevated temperatures and presumably an indication of a thermodynamically stabilized structure. I realize that this may be a covalently coupled array but whatever the product it would shed some light on the relative stability of the 0.4K structure. If the RT structure is not possible to study for some reason then it would be useful to know what happens if the structure is heated at all. Is the unusual tessellation lost?

The phrase "Considering that the herringbone reconstruction disappears in the β phase" should be reworded – disappears is a misleading word, this isn't a magic trick.

The authors also miss some highly relevant papers including a recent review of complexity in 2D structures (Chem. Commun., 2017, 53, 11528) and structures formed by self-assembly through hydrogen bonding (Science, 2008, 322, 1077; Nature Chem., 2012, 4, 112) which should be mentioned as they came before the many of the papers they do cite.

Reviewer #2 (Remarks to the Author):

F. Cheng and co-authors report an original study of tessellation in two dimensions using a combination of halogen-halogen and halogen-metal bonds.

This work is novel and exciting and overall very suitable for the readership of Nature Communications.

I recommend acceptance subject to some minor revision, as follows.

The English could use some polishing. Some expressions such as "It is should be noted that..." besides incorrect (in this case) are unnecessary.

Also "one should note that" and "self-assembly have been reported"

Other than this, I suggest adding a few key references; in particular, on page 4, before results and discussion, the authors refer to the famous interplay between intermolecular and molecule-substrate interactions. This should be referenced. Since the concept is relatively well known, it is sufficient to cite one or two review articles.

In addition, while non-covalent interactions such as Hydrogen bonding, van der Waals forces and metal organic coordination have been used in surface science studies for a long time, halogen bonding in two dimensions is fairly recent. As such the authors should reference the first studies on this topic, specifically

Chem. Comm. 47, 9453–9455 (2011);

Nanoscale 4, 5965–5971 (2012)

Finally, the conclusions are too dry. There is little outlook, perhaps only the last sentence. The authors should make an effort to provide perspectives for future work.

Reviewer #3 (Remarks to the Author):

Prof. Loh and co-workers report the on-surface design of supramolecular architectures of increasing degree of complexity based on a competition between halogen-halogen, halogen-metal and metal-substrate interaction.

The main result of their work is the formation of unconventional higher-order tessellations by such a competition of interactions.

The results are sound, the manuscript is well written and the methodology is appropriate. Thus, I would recommend publication, provided my comments are addressed.

MAJOR COMMENTS:

- 1.- A proper study of the dependence of the distinct phases with respect to coverage should be scholarly introduced and reported.
2. - In figure 4, an atomistic schematic model should be provided to identify molecule, halogen and Au adatom.
- 3.- The authors mention interweaving, but in my modest opinion I think it would be better to describe this phenomenon as a competition of interactions, leading to distinct phases, even though three of them could be rationalized as complex tessellations.

MINOR COMMENTS:

- 2.- Quasicrystals are also used for photonic crystals. This issue should be amended in the introduction.
- 3.- Page 4: Typo. 'It should' instead of 'It is should'
- 4.- Are the STS acquired with the same tip? It is very strange not to see the fingerprint of the surface

state, even without the presence of the herringbone reconstruction.

5.- Where are the frontier orbitals located experimentally, in order to compare with their DFT results?

6.- How are the theoretical images simulated? How do they compare with standard constant current STM images, where a convolution between topography and density of states is always present?

Point-by-point response to “Title: Two-Dimensional Tessellation by Molecular Tiles Constructed from Halogen-Halogen and Halogen-Metal Networks”

Reviewer #1 (Remarks to the Author):

Comment 1: *This is a very nice piece of work with some beautiful structures which are described in a clear manner. Such complex structures showing periodic tessellations of different tiles are rare and of significant interest. As the authors note the controlled assembly of such structures should have interest for photonic materials. Of course the applications are beyond the scope of this paper but it is important to develop the fundamental underpinning science. The authors use a combination of high quality scanning probe studies supported by calculations to provide an evaluation of the observed structures.*

Overall I recommend that the results are of significant interest but that a number of points need to be considered and answered.

Response: We thank reviewer for the high evaluation of our work.

Comment 2: *The authors comment that “The phase evolution of our supramolecular network is convoluted with the coverage-dependent evolution of surface stress on the Au(111) substrate. A detailed investigation is beyond the scope of the current study,”*

Although I agree that there is a large amount of work to be able to determine a full phase diagram for the system and that is indeed beyond the scope of this study. However it would be good if the authors could clarify some points. The molecules are deposited onto the substrate at room temperature, presumably 298K. What is the structure of the assembly at that temperature, prior to cooling to 0.4 K? This should be an easy experiment to do, if the authors haven't already done it, and this would give an indication of the structure at elevated temperatures and presumably an indication of a thermodynamically stabilized structure. I realize that this may be a covalently coupled array but whatever the product it would shed some light on the relative stability of the 0.4K structure. If the RT structure is not possible to study for some reason then it would be useful to know what happens if the structure is heated at all. Is the unusual tessellation lost?

Response: We have actually imaged the sample at room temperature but did not include the data at first as imaging at room temperature does not provide a clear image due to thermal drift. As shown in Figure R1, the α and β phases are thermodynamically stable at room temperature since they can be observed at 298K. Although the image quality is not as good as that at 0.4K, the measured lattice constants and the appearance of Au(111) herringbone reconstruction confirmed that the observed phases are α and β phases. When we scanned different areas of sample, structures A, B and C were not detected, which indicates these higher-order tessellations are not as stable as α and β phases. However, we should note that the scan rate was increased by 5 times at 298K compared with that at 0.4K in order to overcome the thermal drift, thus it is entirely plausible that the fast scan rate disturbs the assembly at 298K. Indeed, we see some fluctuating features (black arrows in Figure R1a) on surface, which is possibly due to the tip induced fast molecular movement.

Figure R1. STM image of HPBI on Au(111) measured at 298K. a, Large-scale STM image. The black arrows mark the area with dynamic behavior. b, Magnified STM image of α phase. The white arrows highlight Au(111) herringbone structure. c, Magnified STM image of β phase.

Comment 3: *The phrase “Considering that the herringbone reconstruction disappears in the β phase” should be reworded – disappears is a misleading word, this isn’t a magic trick.*

Response: Following reviewer’s suggestion, the following sentence has been changed.

Page 5, 2nd paragraph, “Considering that the herringbone reconstruction ~~disappears~~ is lifted in the β phase”

Comment 4: *The authors also miss some highly relevant papers including a recent review of complexity in 2D structures (Chem. Commun., 2017, 53, 11528) and structures formed by self-*

assembly through hydrogen bonding (Science, 2008, 322, 1077; Nature Chem., 2012, 4, 112) which should be mentioned as they came before the many of the papers they do cite.

Response: We are sorry about missing the important references. Actually, the science paper was cited as reference 30 (page 3, paragraph 2), and the rest two was missed out. We have cited them as reference 33 and 34 in the following sentence.

Page 3, 2nd paragraph, “In the past decade, great effort has been devoted to the development of semi-regular AT²³⁻²⁷ and complex tilings²⁸⁻³⁴ on surfaces using self-assembly approaches.”

Reviewer #2 (Remarks to the Author):

Comment 1: *F. Cheng and co-authors report an original study of tessellation in two dimensions using a combination of halogen-halogen and halogen-metal bonds.*

This work is novel and exciting and overall very suitable for the readership of Nature Communications.

I recommend acceptance subject to some minor revision, as follows.

Response: We thank the reviewer for the positive comments on our work.

Comment 2: *The English could use some polishing. Some expressions such as "It is should be noted that..." besides incorrect (in this case) are unnecessary. Also "one should note that" and "self-assembly have been reported"*

Response: Thanks for correcting English error. We have changed accordingly.

Page 4, 3rd paragraph, “~~It is should be noted that a~~ A herringbone-like reconstruction is clearly visible in the α phase (white arrows in Fig. 1c), but is not detected in β phase, which indicates different molecule-substrate interactions in the two phases.”

Page 9, 1st paragraph, “~~One should note that the~~ The red triangle is perfectly aligned with one of the blue triangles (highlighted by dashed white lines), leading to 60° and 120° interior angles in the parallelogram.”

Page 5, 2nd paragraph, “Previous studies have reported that the removal of the herringbone reconstruction is accompanied by the appearance of Au adatoms,^{49,50} and the formation of Au adatoms-mediated self-assembly⁴⁹⁻⁵¹. Considering that the herringbone

reconstruction ~~disappears-is lifted~~ in the β phase, it is reasonable to propose that the molecular network is stabilized by coordination between I atoms and Au adatoms. ~~Au adatoms-mediated molecular self-assembly have been reported previously⁴⁹⁻⁵¹.~~

Comment 3: *Other than this, I suggest adding a few key references; in particular, on page 4, before results and discussion, the authors refer to the famous interplay between intermolecular and molecule-substrate interactions. This should be referenced. Since the concept is relatively well known, it is sufficient to cite one or two review articles. In addition, while non-covalent interactions such as Hydrogen bonding, van der Waals forces and metal organic coordination have been used in surface science studies for a long time, halogen bonding in two dimensions is fairly recent. As such the authors should reference the first studies on this topic, specifically Chem. Comm. 47, 9453–9455 (2011); Nanoscale 4, 5965–5971 (2012)*

Response: Thanks for suggestion these references. Following the suggestion, we have added the reference 40 and 41 for “halogen bond^{40,41}”, and reference 42 and 43 for “interplay between intermolecular and molecule-substrate interactions^{42,43}” (page 4, 2nd paragraph).

Comment 4: *Finally, the conclusions are too dry. There is little outlook, perhaps only the last sentence. The authors should make an effort to provide perspectives for future work.*

Response: Following reviewer’s suggestion, we have rewritten the conclusion and cited the related paper, reference 64.

Page 11, 2nd paragraph, “The self-similarity of the two molecular phases allow clusters from each to tessellate and form high-order supramolecular networks. This approach opens up new routes to construct complex surface tessellation by considering the symmetry of building block and substrate, as well as introducing multimode interactions. In addition, the complex tessellations in our work may provide new insights for understanding self-organized systems in biology and nanotechnology⁶⁴.”

Reviewer #3 (Remarks to the Author):

Comment 1: *Prof. Loh and co-workers report the on-surface design of supramolecular architectures of increasing degree of complexity based on a competition between halogen-halogen, halogen-metal and metal-substrate interaction. The main result of their work is the formation of unconventional higher-order tessellations by such a competition of interactions.*

The results are sound, the manuscript is well written and the methodology is appropriate. Thus, I would recommend publication, provided my comments are addressed.

Response: We thank reviewer for the positive comments on the quality of our research and the presentation in our manuscript. We will reply to all the comments in a point-by point fashion below.

MAJOR COMMENTS:

Comment 2: *1.- A proper study of the dependence of the distinct phases with respect to coverage should be scholarly introduced and reported.*

Response: We summarized the phase evolution under different coverages in Figure R3. At low coverage (<0.40ML, 1ML is defined as one monolayer of HPBI in α phase), only α phase is observed. When the coverage is increased to about 0.50ML, β phase emerges and coexists with α phase. Further increasing the coverage, some disordered mix phase, and the ordered structure A, B, C appear on surface. We tried even higher coverage, but the proportion of the ordered structure A, B, C did not show much change. We think these ordered tessellations may be very sensitive to the thermodynamics and kinetics factors. To prepare large area of these phases may require dedicated tuning of growth conditions (such as, flux ratio, substrate temperature, sample cooling rate) or even choose different substrates. This can be investigated in future work.

Following reviewer's suggestion, we included Figure R3 as supplementary Figure 4 and clarify the coverage dependence in revised supplementary information, "At low coverage, only α phase is observed. When the coverage is increased to about 0.50ML, β phase emerges and coexists with α phase. Further increasing the coverage, some disorder mix phase, and the ordered structure A, B, C appear on surface. (Page 10, figure caption)".

In the revised manuscript, we have made following changes to describe phase evolution at high coverage. (The coverage dependence for α and β phases at low coverage was discussed in page 5, 3rd paragraph).

Page 8, 1st paragraph, “In this work, in addition to the large scale, phase-separated α and β phases, inhomogeneously mixed phases (Supplementary Fig. 4 and Supplementary Fig. 5) are indeed observed in select local areas as the HPBI coverage was increased above 0.55ML (1ML is defined as one monolayer of HPBI in α phase). Most intriguingly, three types of ordered mixed-phase structures (named A, B, and C) are discovered under high coverage regime ~~herein~~; STM images and corresponding FFT patterns of these images show periodic arrangements of close-packed clusters with six-fold symmetry, as shown in Fig. 4b-d and insets therein.”

Figure R3. STM image of Au(111) after deposition of HPBI at 0.02ML (a), 0.40ML (b), 0.50ML (c), 0.55ML (d) and 0.60ML (e). The different phases are labelled in images.

Comment 3: 2.- *In figure 4, an atomistic schematic model should be provided to identify molecule, halogen and Au adatom.*

Response: We thank the reviewer for this suggestion. For revision, the molecule, I-atoms and Au adatoms are marked, based on the style we employed previously for Figure 1. In addition, Figure 4g-j have been rotated for the better comparison with Figure 4b-d. The revised Figure 4 is shown here, and the figure caption has been adjusted.

Page 23, 1st paragraph, “**e-g**, High-resolution STM images of structures A (e), B (f), and C (g). The HPBI molecules are labelled by the dashed white and green circles, and the Au adatoms are labelled by blue circles. The interwoven vertices are marked by black and yellow circles.”

Comment 4: 3.- *The authors mention interweaving, but in my modest opinion I think it would be better to describe this phenomenon as a competition of interactions, leading to distinct phases, even though three of them could be rationalized as complex tessellations.*

Response: We agree with the reviewer that the underlying mechanism is due to competition between halogen-halogen, halogen-metal and metal-substrate interaction. The word

“interweaving” is more of a graphical description to indicate the “interpenetration of one pattern into another”. To address this point, we have added the following sentence.

Page 10, 1st paragraph, “The formation of the manifold complex tessellations is most likely due to the competition between halogen-halogen, halogen-metal as well as metal-substrate interaction.”

MINOR COMMENTS:

Comment 5: 2.- *Quasicrystals are also used for photonic crystals. This issue should be ammended in the introduction.*

Response: We are sorry about this mistake. For revision, the corresponding sentence have been adjusted.

Page 3, 2nd paragraph, “Compared with ~~quasicrystal or~~ non-periodic tessellation, ~~quasicrystal or~~ periodic tessellations with high degree of rotational symmetry ~~is—a—are~~ promising ~~candidate-candidates~~ for 2D photonic crystals¹⁰.”

Comment 6: 3.- *Page 4: Typo. 'It should' instead of 'It is should'*

Response: We thank the reviewer for pointing out this typo. To be concise, we have made the following changes.

Page 4, 3rd paragraph, “~~It is should be noted that a~~ herringbone-like reconstruction is clearly visible in the α phase (white arrows in Fig. 1c), but is not detected in β phase, which indicates different molecule-substrate interactions in the two phases.”

Comment 7: 4.- *Are the STS acquired with the same tip? It is very strange not to see the fingerprint of the surface state, even without the presence of the herringbone reconstruction.*

Response: Indeed the same tip was used for measuring STS on α and β phases. Moreover, before measurements on molecules, the tip was tested on bare Au(111) area and the characteristic Au surface state was imaged. The same check was also conducted after STS measurements on molecules. Moreover, STS in bigger bias range (lower panel of Figure R4b) also does not show the surface state. Therefore, we believe our data is trustworthy.

We agree with the reviewer that the surface state may still exist even when herringbone reconstruction is lifted. However, our STS were conducted on the molecules, and not directly on Au. Previously, Pia and co-workers (Nanoscale, 2016, 8, 19004) found that Au(111) surface state disappeared in STS for self-assembly layer of 7,7,8,8-tetracyanoquinodimethane (TCNQ) stabilized by TCNQ-Au adatom coordination. Gerbert (Phys. Rev. B 2017, 96, 144304) reported the disappearance of Au(111) surface state by STS for tetrathiafulvalene (TTF)-covered Au(111) surface. The disappearance of surface state was attributed to significant structure change or charge rearrangement at metal-organic interface. The disappearance of surface state in our case is possibly due to the same reason.

In order to avoid misunderstanding, we have made the following changes and cited the above papers as reference 58 and 59.

Page 6, 2nd paragraph, “However, STS spectra of the β phase molecules show no evidence of the Au(111) surface state, ~~suggesting that the surface state is completely quenched. This observation indicates that the underlying Au(111) structure has been altered~~ this may be due to structural change or charge rearrangement^{58,59} at the Au-HPBI interface in the β phase, which is consistent with the appearance of Au adatoms on the substrate. ”

Comment 8: 5.- *Where are the frontier orbitals located experimentally, in order to compare with their DFT results?*

Response: To compare with calculated frontier orbitals, STS were measured locally for the α and β phase. Figure R4b shows a peak located at -1.6V in α phase, and a peak at -1.8V in β phase. Moreover, Au(111) surface state can be detected in α phase but not in β phase. This overall features in STS is similar to the calculated results (Figure R6a). However, the exact energy positions (-1.6V vs. -1.1V, -1.8V vs. -1.5V) are different in experiments and calculations. Several factors may cause the discrepancy, such as, the calculation was performed on Gamma point, but STS give the local density of states (LDOS) in Brillouin zone; the image charge was not considered in calculations.

Figure R4. (a) Projected density of states of I atoms for α and β phases. (b) STS acquired at different locations of molecule in α and β phases.

The purpose for calculating frontier orbitals was to study the intermolecular interaction via I-atoms. Therefore, only the projection on I-atoms was considered. To make the description clearer, the following sentences have been adjusted, and two related references have been added.

Page 7, 1st paragraph, “For the α phase, the highest occupied molecular orbitals (HOMO) and lowest unoccupied molecular orbitals (LUMO) have their projections appear in-on the I atoms at -1.1 eV and 2.3 eV, respectively (upper panel of Supplementary Fig. 1).”

Page 7, 1st paragraph, “On the other hand, the distances between adjacent I atoms are much larger in the β phase due to coordination with Au adatoms to form I–Au–I interactions, as indicated by the plotted wavefunctions of ~~shown in the I atom~~ HOMO (-1.5 V) and LUMO (2.9 V) ~~state wave functions~~ shown in Fig. 3f. The three-center orbital hybridization has been also reported in N–Au–N and biphenyl trimer systems previously^{60, 61}.”

Page 20, figure caption, “**e,f**, Real-space visualized wavefunctions of HOMO and LUMO ~~projections of the states of the I atom frontier orbitals~~ in the α (e) and β (f) phases.”

Comment 9: 6.- How are the theoretical images simulated? How do they compared with standard constant current STM images, where a convolution between topography and density of states is always present?

Response: The DFT simulation for STM images was conducted using the Tersoff-Hamann approximation (reference 71, Phys. Rev. B 1985, 31, 805). For a two-dimensional surface, the charge densities within a specific energy range, which are selected to match with the

experimental bias, are calculated pixel by pixel. The constant charge density contour is generated as the simulated constant-current image, where the intensity of the specific pixel is determined by its vertical height. Experimentally, the constant-current STM image is acquired by scanning tip across the surface, and employing a feedback loop to keep the tunnelling current constant through controlling z piezo; the trajectory of tip is used to generate STM image. According to Tersoff-Hamann approximation, the tunnelling current at a small bias is proportional to LDOS near Fermi level. Therefore, STM image can be understood as a contour map of LDOS of surface. This is what calculated in the simulated STM image. Since topographic and electronic effects are two indivisible parts, both are considered in the simulation. The interaction between tip and surface is ignored in the calculations. To be clear, we have included the following description in the method part.

Page 12, 3rd paragraph, “STM simulation was performed using the Tersoff-Hamann approximation⁷⁰. The charge densities on surface within a specific energy range, which are selected to match with the experimental bias, are calculated pixel by pixel. The constant charge density contour is generated as the simulated constant-current image, where the intensity of a specific pixel is determined by its vertical height. The interaction between tip and surface is ignored in the calculations.”

REVIEWERS' COMMENTS:

Reviewer #1 (Remarks to the Author):

In my opinion the authors have addressed all the comments raised by the referees and I recommend publication without further change.

Reviewer #3 (Remarks to the Author):

The authors have satisfactorily addressed all comments from referees.

Therefore, I would strongly recommend publication in Nature Communications.